# LABELED TRUSTSET GUIDED: COMBINING BATCH ACTIVE LEARNING WITH REINFORCEMENT LEARNING

## ABSTRACT

Batch active learning (BAL) is a crucial technique for reducing labeling costs and improving data efficiency in training large-scale deep learning models. Traditional BAL methods often rely on metrics like Mahalanobis Distance to balance uncertainty and diversity when selecting data for annotation. However, these methods predominantly focus on the distribution of unlabeled data and fail to leverage feedback from labeled data or the model's performance. To address these limitations, we introduce TrustSet, a novel approach that selects the most informative data from the labeled dataset, ensuring a balanced class distribution to mitigate the long-tail problem. Unlike CoreSet, which focuses on maintaining the overall data distribution, TrustSet optimizes the model's performance by pruning redundant data and using label information to refine the selection process. To extend the benefits of TrustSet to the unlabeled pool, we propose a reinforcement learning (RL)-based sampling policy that approximates the selection of high-quality TrustSet candidates from the unlabeled data. Combining TrustSet and RL, we introduce the **B**atch **R**einforcement **A**ctive **L**earning with **T**rustSet (**BRAL-T**) framework. BRAL-T achieves state-of-the-art results across 10 image classification benchmarks and 2 active fine-tuning tasks, demonstrating its effectiveness and efficiency in various domains.

## 1 INTRODUCTION

In the era of deep learning, large-scale labeled datasets are indispensable for training models on complex tasks. Active learning (AL) provides an efficient approach to reduce the labeling costs by intelligently selecting critical subsets from unlabeled data for annotation (Zhan et al., 2022). Batch active learning (BAL) (Citovsky et al., 2021), a variant of AL, further improves this process by selecting data points in groups (batches), thereby reducing the overhead associated with model retraining and oracle interactions.

In most modern BAL methods, the selection strategy is typically based on two factors: uncertainty and diversity. Uncertainty-based methods focus on choosing the most ambiguous or difficult data, which is likely to improve the model, but this often results in selecting redundant data that doesn't sufficiently cover the data distribution (Shen et al., 2017). On the other hand, diversity-based methods aim to ensure a representative subset by covering as many different types of data as possible, but they may neglect critical uncertain samples near the decision boundaries. For instance, CoreSet (Phillips, 2017) selects subsets that reflect the overall data distribution, ensuring diversity by minimizing the distance between the selected subset and the full dataset. While methods like Cluster-Margin(Citovsky et al., 2021) combine diversity and uncertainty to improve data selection, they still have limitations, such as overlooking feedback from the labeled dataset, ignoring class distribution, and potentially inheriting the long-tail distribution problem.

To address these challenges, we propose TrustSet, a novel data selection approach that distinguishes itself from CoreSet by emphasizing the utilization of label information. TrustSet focuses not only on ensuring diversity but also on selecting data that is most beneficial for improving the model's performance, especially in cases where class imbalance is a concern. TrustSet differs from CoreSet in two significant ways:

**Objective**: TrustSet is designed to optimize the model's performance, with an explicit focus on improving accuracy and tackling the long-tail distribution problem by selecting crucial data that has a high potential to be forgotten by the model (Toneva et al., 2018). In contrast, CoreSet focuses on representing the full data distribution without directly considering the impact on the model's learning process, which can lead to inheriting undesirable distributional imbalances.

**Data Source**: TrustSet leverages labeled data, utilizing ground truth labels to prune redundant and noisy data and ensure that the selected subset is balanced across classes. This approach contrasts with CoreSet, which selects data purely from the unlabeled pool, without considering feedback from the trained model. As a result, CoreSet-based methods can miss the opportunity to incorporate critical information about the model's current performance, potentially leading to suboptimal data selections.

TrustSet's balanced class distribution ensures better handling of the long-tail distribution problem, where underrepresented classes are more likely to be included in the training process. To construct TrustSet, we use the GradNd method (Paul et al., 2021), which ranks data based on the gradient norms of model updates, prioritizing data points that contribute most to model learning. Furthermore, to improve the data quality, we incorporate SuperLoss (Castells et al., 2020), which follows a curriculum learning strategy to assign higher importance to easier data in early training stages, while still considering difficult samples later.

However, extending the TrustSet concept to the unlabeled data pool presents a challenge, as the selection process requires label information. To overcome this, we introduce an RL-based policy for approximating the selection of high-potential TrustSet candidates from the unlabeled data pool. Unlike previous RL-based active learning approaches, which often require frequent retraining of the model and rely heavily on complex reward structures (Fang et al., 2017; Zhang et al., 2023), our method minimizes retraining costs by leveraging TrustSet to guide the reinforcement learning process.

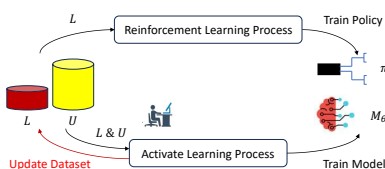

Figure 1: Overview of the BRAL-T framework.

To this end, we propose a novel batch active learning framework called BRAL-T (**B**atch **R**einforcement **A**ctive **L**earning with **T**rustSet extraction), which integrates TrustSet and RL-based policies for efficient data selection. The framework consists of two primary components: (1) TrustSet extraction, which ensures that the labeled dataset contributes optimally to model performance and maintains a balanced class distribution, and (2) RL-based subset selection, where a learned policy selects from the unlabeled data pool to approximate TrustSet. This significantly reduces the need for repeated oracle queries and model retraining. As shown in Figure 1, BRAL-T is implemented with two processes: reinforcement learning (RL) for policy training and active learning (AL) for model training.

Our contributions are summarized as follows:

- We introduce TrustSet, a novel method for data selection that leverages label information to balance uncertainty, diversity, and class distribution, thus addressing the long-tail distribution problem.
- We develop an RL-based data selection policy that bridges the gap between TrustSet's label dependency and the unlabeled setting of active learning, allowing for more efficient and targeted data selection.
- We propose BRAL-T, a new batch active learning framework that integrates TrustSet and RL to reduce the computational burden of active learning while improving model performance. We demonstrate that BRAL-T achieves state-of-the-art performance across multiple image classification and active fine-tuning tasks.

## 2 RELATED WORK

**Active Learning:** Active learning, vital for reducing labeling costs, focuses on extracting insights from unlabeled dataset features and using models trained on labeled data for data selection. This

field hinges on uncertainty and diversity. (Shen et al., 2017) investigated uncertainty-based methods in active learning for Named Entity Recognition (NER), later integrating diversity for enhanced outcomes. Galaxy ((Zhang et al., 2022)) emphasized uncertainty by constructing a model confidence graph, with the median node indicating high uncertainty. Conversely, (Yuan et al., 2020) prioritized diversity by using self-supervised models to represent datasets in an embedded feature space, selecting central points from clusters for broad dataset coverage, thus illustrating diversity's role in active learning. Recent studies ((Liu et al., 2019; Ash et al., 2019; Sinha et al., 2019; Margatina et al., 2021; Ash et al., 2021; Citovsky et al., 2021; Kim et al., 2021; Gentile et al., 2022)) highlight a trade-off in active learning between uncertainty and diversity, with uncertainty-based subsets often being redundant and not fully representative, while diversity-focused subsets may miss critical uncertain data. Current research seeks to balance these aspects. However, traditional batch sampling methods in active learning tend to ignore the distribution of selected data, leading to further redundancy.

**Batch Active Learning:** Active learning methods, aiming to minimize oracle queries, prefer batch data processing over individual sample handling. Batch active learning approaches ((Zhang et al., 2023; Ash et al., 2021; Kirsch et al., 2019; Citovsky et al., 2021; Sener & Savarese, 2017)) concentrate on batch sampling to reduce costs and preserve subset distribution. BatchBald ((Kirsch et al., 2019)) highlighted that batch applications of single-data methods often lack diversity and joint informativeness, and proposed an iterative entropy-based batch sampling for enhanced information gain. Cluster-Margin ((Citovsky et al., 2021)) employed Hierarchical Agglomerative Clustering (HAC) for one-time clustering of the unlabeled data pool, focusing on the most uncertain data, and supported scaling up to batches of 1M. Despite their effectiveness in computer vision tasks, these methods overlook feedback from the selected subset, such as accuracy changes, which could be crucial for refining data sampling strategies.

**Active Learning with RL:** Rather than designing sample strategies based on expert knowledge, several active learning methods apply RL to learn sample policies based on the performance of the selected subset ((Zhang et al., 2023; Fang et al., 2017; Liu et al., 2019; Gong et al., 2022; Smit et al., 2021; Casanova et al., 2020)). Some methods ((Fang et al., 2017; Gong et al., 2022; Smit et al., 2021; Casanova et al., 2020)) defined rewards as evaluation results of the trained target model, such as accuracy, AUROC or prediction change. However, frequent retraining of the target model is required to collect enough reward data for RL training and the credit assignment problem exists. Instead, (Liu et al., 2019) defined the reward as the Mahalanobis distance between selected data and existing labeled data. TAILOR ((Zhang et al., 2023)) maintained the class distribution of candidate active learning methods and formulated class balance as the reward for a contextual bandit problem. Both methods reduce the difficulty of RL training, but the method of (Liu et al., 2019) was designed specifically for the Re-ID task, and the reward defined by TAILOR ((Zhang et al., 2023)) did not directly correlate with the accuracy of the target model. As a result, we propose an RL-based active learning method without requirement of target model retraining and has a high correlation with target task (e.g., classification accuracy).

## 3 PROBLEM DEFINITION

In this section, we formally define active learning problem in batch setting following (Sener & Savarese, 2017) and TrustSet selection problem. We are interested in a $C$-class classification task over a compact space $\mathcal{X}$ and a label space $\mathcal{Y} = \{1, \ldots, C\}$. We aim to train a target model $M_\theta$ with parameter $\theta$ to optimize a loss function $l(\cdot, \cdot; M_\theta) : \mathcal{X} \times \mathcal{Y} \to R$. In practice, we consider a large collection of data points sampled *i.i.d* over the space $\mathcal{Z} = \mathcal{X} \times \mathcal{Y}$ as $\{\mathbf{x}_k, y_k\}_{k \in [n]} \sim p_{\mathcal{Z}}$.

For active learning problem, we further define labeled dataset with $|L|$ data points as $L = \{\mathbf{x}_k, y_k\}_{k \in |L|}$ and unlabeled data pool with $|U|$ data points as $U = \{\mathbf{x}_k\}_{k \in |U|}$. In general, $|L| \ll |U|$ and $L \bigcap U = \emptyset$. We aim to select a data subset $S \sim U$ which will be labeled by an oracle and applied to enhance labeled dataset $L$. The model $M_\theta$ trained on the enhance labeled dataset is expected to have better performance. Thus, active learning problem can be formulated as follow:

$$S_i = \underset{S \subseteq U_i : |S| \leq b}{\arg\min} E_{\mathbf{x}, y \sim p_{\mathcal{Z}}}[l(M_{\theta_{L_{i+1}}}(\mathbf{x}), y)] \tag{1}$$

where $S_i$ refers to the selected subset and $L_{i+1} = L_i \bigcup S$ indicates the enhance labeled dataset in the $i$th active learning iteration. $M_{\theta_{L_{i+1}}}$ refers to model $M_\theta$ trained on labeled dataset $L_{i+1}$ and $b$

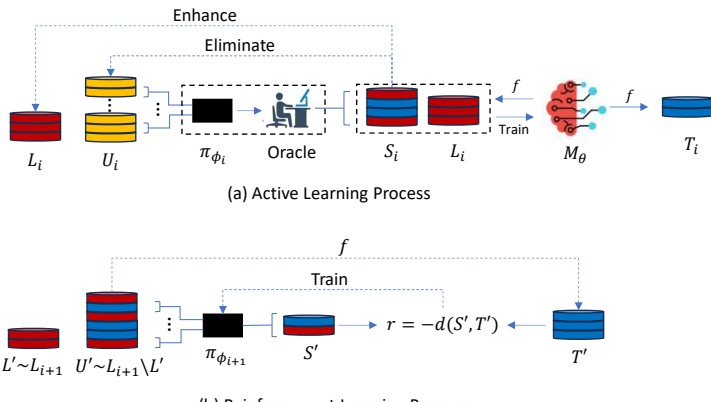

Figure 2: Details of BRAL-T. (a) In the active learning process, policy $\pi_{\phi_i}$ selects the subset $S_i$ from $U_i$ for the oracle to annotate, and the model $M_{\theta_i}$ is trained on $S_i \cup L_i$. (b) In the RL process, we sample $L'$ and $U'$ from $L_{i+1}$ to train policy $\pi_{\phi_{i+1}}$. The reward function is defined to encourage similarity between $S'$ and $T'$.

refers to the size of data subset selected in each iteration. As a result, Eq. 1 indicates that we aim to select a data subset $S_i$ from $U_i$ to enhance $L_i$ such that trained model $M_{\theta_{L_{i+1}}}$ achieves minimal loss value.

Directly solving above optimization problem is challenge due to the lack of label information from $U_i$. So we propose to analyze the labeled dataset $L_i$ instead to discover most important data that contribute to performance improvement of trained model. We formally define TrustSet $T_{L_i}$ as follow:

$$T_{L_i} = \underset{S \subseteq L_i : |S| \leq b'}{\arg\min} \ E_{\mathbf{x}, y \sim p_{\mathcal{Z}}}[l(M_{\theta_S}(\mathbf{x}), y)] \quad \text{s.t. balance}(S) \tag{2}$$

which indicates that $T_{L_i}$ refers to a data subset $S$ from labeled dataset $L_i$ that could optimize the performance of trained model $M_{\theta_S}$. And we require $S$ to be balanced to alleviate long-tail problem. The optimization problem of Eq. 2 is similar to that of Eq. 1, but the accessible of label information from $L_i$ brings possible solution of Eq. 2. TrustSet $T_{U_i}$ also exists in $U_i$ and we redefine active learning problem based on TrustSet as:

$$S_i = \underset{S \subseteq U_i, |S| \leq b}{\arg\min} \ d(S, T_{U_i}) \tag{3}$$

where $d$ refers to statistical distance between two datasets. However, it is impossible to train model $M_{\theta_S}$ without label on $U_i$ and optimize the loss function in Eq. 2 to extract $T_{U_i}$.

To solve this problem, we train a data selection policy $\pi_{\phi_i}$ with RL method in labeled dataset $L_i$ learning to select $S_i$ and approximate $T_{U_i}$. We create a similar environment as active learning setting by randomly sampling a data subset $L'$ with labels and another subset $U'$ without label information from the existed labeled dataset $L_i$. We ensure $L' \bigcap U' = \emptyset$ and $|L'| \ll |U'|$. Although we omit the labeled information of $U'$ for $\pi_{\phi_i}$ input, we still can leverage label of $U'$ to extract $T'$, TrustSet of $U'$, based on Eq. 2. Thus, suppose $\pi_{\phi_i}$ takes $L'$ and $U'$ as input to select data subset $S'$ from $U'$, we define the reward function for training policy $\pi_{\phi_i}$ as:

$$R = -d(S', T') \tag{4}$$

where $S' = \pi_{\phi_i}(L', U')$ and we optimize parameters $\phi_i$ to minimize the statistical distance between $T'$ and $S'$. In this way, $\pi_{\phi_i}$ learns to select data from unlabeled dataset with high potential to be included in TrustSet based on feature space. After training, $\pi_{\phi_i}$ will be applied to the real unlabeled data pool $U_i$ to select $S_i = \pi_{\phi_i}(L_i, U_i)$.

In general, for each active learning iteration, we solve Eq. 1 in two processes as shown in Figure 2. **In the active learning process**, data subset $S_i$ is selected by $\pi_\phi(L_i, U_i)$ and passed for oracle to annotate. $M_\theta$ is then trained on the enhanced labeled dataset $L_{i+1} = L_i \bigcup S_i$. Moreover, new unlabeled data pool $U_{i+1}$ is achieved by eliminate $S_i$ from $U_i$ and TrustSet $T_i$ is extracted from

$L_{i+1}$ by solving Eq. 2 for the convenience of next RL process. **In the RL process**, we create the environment based on $L_{i+1}$ and train $\pi_{\phi_{i+1}}$ with reward function as Eq. 4.

## 4  METHOD

In this section, we introduce BRAL-T framework in detail. In Section 4.1, we introduce a TrustSet construction method based on GradNd score ((Paul et al., 2021)). In Section 4.2, we illustrate the details of the RL module and describe how we use the learned policy to select subsets from the unlabeled pool.

### 4.1  TRUSTSET

In general, the TrustSet should retain important data and tend to be class-balanced. However, it is almost impossible to solve Eq. 2 due to the large size of labeled dataset and time-consuming of $M_{\theta_S}$ training. As a result, we introduce a TrustSet extraction method based on the GradNd score ((Paul et al., 2021)) by analyzing the performance of model trained on entire trainset $L$ rather than selected subset $S$. This score is defined as the expected value of the gradient norm term with respect to a differentiable model and a data sample $x$:

$$GradNd = E\| \sum_{k=1}^{K} \nabla_{M_\theta^{(k)}} \ell(M_\theta(x), y)^T \nabla_\theta M_\theta^{(k)}(x)\| \tag{5}$$

In this equation, $\ell$ refers to the loss function, and $y$ is the label of the corresponding data sample $x$; $K$ denotes the number of logits, and $M_\theta^{(k)}(x)$ represents the result of the $k$-th logit. For instance, in an image classification task, $\ell$ represents the cross-entropy loss, $K$ is the number of categories, and $M_\theta^{(k)}(x)$ is the logit output for the $k$-th category. Data samples that result in a large gradient value tend to contain information that the model has not yet learned, as the model would update significantly based on such data. As demonstrated by the experimental analysis from (Paul et al., 2021), data with a higher GradNd score tend to be forgotten samples for the target model during training and are more important for further training. However, the GradNd score might lead to a class imbalance problem when the data subset primarily contains difficult images for certain categories. To mitigate the long-tail distribution problem, we sort data by class using the GradNd score and select the top-N data for each category. For the image classification task, we follow (Paul et al., 2021) in omitting the term $\nabla_\theta M_\theta^{(k)}(x)$ from Eq 5 and calculate the approximated EL2N score.

**Curriculum Learning:** Data with high GradNd scores tend to be difficult and uncertain samples. As suggested by previous works ((Ash et al., 2021; Citovsky et al., 2021; Gentile et al., 2022)), a training set focusing on uncertainty could result in high redundancy and fail to train a model that captures general features. We reconsider this issue from another important perspective. Difficult samples contain noise that can interfere with model predictions and increase the difficulty for the model to learn the boundaries between categories. With a limited amount of data, easy examples could help the model capture features and cluster data within the same category. Following the principles of curriculum learning ((Tang & Huang, 2019; Castells et al., 2020)), we assign larger weights to easier data samples in the early active learning iterations and leverage Super Loss ((Castells et al., 2020)) on top of the task loss $\ell_t$. For each data sample $(x, y)$, the super loss $\ell_s$ is defined as:

$$\ell_s(M_\theta(x), y) = (\ell_t(M_\theta(x), y) - \tau)\sigma + \lambda(\log \sigma)^2 \tag{6}$$

where $\tau$ is the threshold for separating easy and hard samples, and $\lambda$ is the weight of the regularization term. Both $\tau$ and $\lambda$ are hyperparameters, while $\sigma$ is learnable and indicates the weight assigned to the task loss. To minimize $\ell_s$, data with task loss $\ell_t < \tau$ will be assigned a larger weight $\sigma$, and data with $\ell_t > \tau$ will be assigned a smaller weight $\sigma$. Since model training on uncertain data typically results in larger losses compared to easier data, super loss adaptively adjusts the weight for data samples. Meanwhile, the scale of $\sigma$ is determined by $\lambda$. As $\lambda$ increases, the value of $\sigma$ tends to be 1 and has less effect on the task loss. Specifically, when $\lambda \to \infty$, $\sigma$ will always be 1 to minimize the regularization term, making $\ell_s$ equal to $\ell_t - \tau$. As shown in Section 5.4, with Super Loss, proposed method achieves better performance.

For convenience, in the following sections, TrustSet $T$ refers to the TrustSet with Super Loss unless explicitly stated.

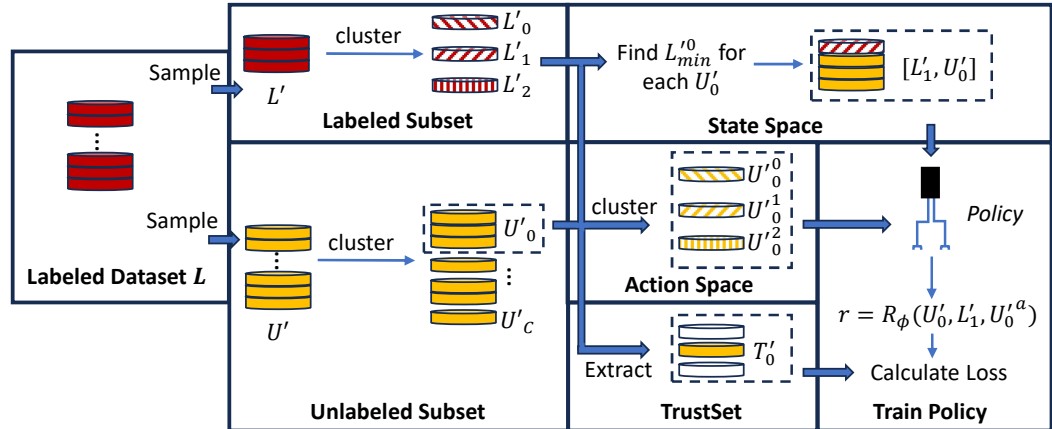

Figure 3: Reinforcement learning process. We cluster $L'$ and $U'$ into $\{L'_m\}_{m=1}^M$ and $\{U'_c\}_{c=1}^C$ as state space, and further cluster each $U'_c$ into $\{U'^a_c\}_{a=1}^{A_c}$ as action space. Given state, action pair as input, we train a Q function to predict distance between $U'^a_c$ and related TrustSet. We use $U'_0$ as an example in the figure.

## 4.2 REINFORCEMENT LEARNING

TrustSet is collected with label information to ensure class balance and better reliability. However, during the active learning process, the policy needs to be applied to select a subset from the unlabeled data pool $U_i$. As a result, we create an environment with similar conditions and apply reinforcement learning to train a policy for subset selection, where the TrustSet is the target subset. In the remainder of this section, we first define the state, action, and reward for the reinforcement learning task in general, and then illustrate the overall process.

**State:** We randomly sample $L'$ as a labeled dataset and $U'$ as an unlabeled data pool from $L_{i+1}$ to train policy $\pi_{\theta_{i+1}}$. We also extract $T'$ from $U'$ as the target TrustSet for each $(L', U')$ sample. However, taking all data in $L'$ and $U'$ as input is time-consuming and challenging for learning a good policy, it is beneficial to have alternative representations. Since in classification task, data tend to be clustered based on predicted category in feature space and $T'$ will spread around all clusters, it is more reasonable to predict $T'$ by using clusters as states. Thus, we define the state space as $(L^c_{min}, U_c)$, where $U_c$ refers to the $c$th cluster from the unlabeled data pool $U$, and $L^c_{min}$ is a cluster from the labeled dataset $L$ defined as:

$$L^c_{min} = \arg\min_m d(L_m, U_c) \tag{7}$$

In this equation, $L_m$ refers to the $m$th cluster from $L$, and $L^c_{min}$ is the closest labeled cluster to $U_c$ based on the distance function $d(\cdot, \cdot)$. We use the Wasserstein Distance ((Flamary et al., 2021)) in our reinforcement learning process. For each $(L', U')$ sample, there are $C$ states, where $C$ is the number of clusters in $U'$. For further efficiency, we extract stochastic features of clusters as input for the policy, specifically using mean and variance as $[E[L^c_{min}], Var[L^c_{min}], E[U_c], Var[U_c]]$. Noted that $L^c_{min}$ seems not to be necessarily considered in state space as $T'$ is extracted from $U'$. But $L^c_{min}$ can provide additional category information for $U'$ and we found adding $L^c_{min}$ improves the performance of active learning by experiment.

**Action:** As data in the TrustSet tend to group together by cluster, we further cluster $U_c$ into $A_c$ data groups as $\{U^a_c\}_{a=1}^{A_c}$. Given $(L^c_{min}, U_c)$ as input, the policy selects the top clusters inside $U_c$ with high potential to be included in the TrustSet. As a result, $\{U^a_c\}_{a=1}^{A_c}$ represents the candidate action space for each state, and the union of selected actions will be the final selected data subset $S$.

**Reward:** Since different $U^a_c$ contains a different number of data, for a fixed size of $S$, we need to select a varying number of $U^a_c$. It is more general to define the reward based on $U^a_c$ rather than $S$. We set the reward function as the negative Wasserstein distance between $U^a_c$ and the sub-TrustSet $T_c$ as:

$$R(U^a_c, T_c) = -d(U^a_c, T_c) \tag{8}$$

where $T_c = T' \cap U_c$, and $U_c^a$ closer in distribution to $T_c$ receives a better reward.

We follow the DQN method ((Mnih et al., 2013)) and show the overall RL process in Figure 3. For the dataset $L'$ and $U'$, we pass them through the target model $M_{\theta_i}$ to obtain the feature space. To create the input for the policy, we cluster the features of $L'$ into $M$ clusters as $\{L'_m\}_{m=1}^M$ and the features of $U'$ into $C$ clusters as $\{U'_c\}_{c=1}^C$. To obtain candidate actions, we further cluster $U'_c$ into $A_c$ clusters as $\{U'^a_c\}_{a=1}^{A_c}$. Meanwhile, we extract the TrustSet for each cluster as $T'_c = T' \cap U'_c$. Unlike traditional RL tasks, we consider the future effect in curriculum learning and TrustSet selection and focus only on the next timestep for policy training. As a result, training the Q function is the same as training the reward function $R_\phi$ as:

$$r = R_\phi(U'_c, L'^c_{min}, U'^a_c) \tag{9}$$

where $\phi$ refers to the parameter of the reward function, and it is updated by the Mean Square Error (MSE) loss as:

$$L = E_{(U'_c, U'^a_c)} \| (R_\phi(U'_c, L'^c_{min}, U'^a_c) - R(U'^a_c, T'_c))^2 \| \tag{10}$$

During training, we calculate the reward for all candidate actions and states to optimize the reward function $R_\phi$. During the active learning process, for each unlabeled cluster $U_c$, we predict the reward for all candidate actions $U_c^a$ and pick them from high to low based on the reward score until the fixed size of subset selection is satisfied.

It is worthwhile to note that based on Section 4.1, for each $(L', U')$ sampled from $L$, $M_\theta$ needs to be retrained on $L'$ to extract TrustSet $T'$ from $U'$ since the important data for a model can vary with different labeled training sets. To avoid the time-consuming process of frequent retraining, we approximate the $T'$ extraction by reusing $M_\theta$ trained on $L_i$ in the $i$th active learning iteration for the reason that our general purpose is to enhance $L_i$ based on the performance of $M_{\theta_{L_i}}$ in the $i$th active learning iteration. the only requirement is to retrain the policy from scratch for each active learning iteration. In practice, we extract $T_i$ from the entire labeled set and extract $T'$ by calculating the intersection between $U'$ and $T_i$ as $T' = U' \bigcap T_i$ for efficiency. Please refer to Appendix for more details of reinforcement learning training.

## 5 EXPERIMENT

In this section, we evaluate our proposed BRAL-T method on the image classification task and compare our results with previous active learning baselines, following the experimental settings of (Zhan et al., 2022). Additionally, we also evaluate BRAL-T on the active fine-tuning task ((Xie et al., 2023)) and compare it with the current state-of-the-art method, ActiveFT, in Section 5.3. Finally, we demonstrate the effectiveness of our proposed modules through an ablation study in Section 5.4. More experimental results will be presented in Appendix C.

### 5.1 IMAGE CLASSIFICATION RESULTS

**Datasets:** We evaluated BRAL-T on the image classification task across 8 benchmarks, including Cifar10, Cifar100 ((Krizhevsky et al., 2009)), Cifar10-imb, EMNIST ((Cohen et al., 2017)), FashionMNIST ((Xiao et al., 2017)), BreakHis ((Spanhol et al., 2015)), Pneumonia-MNIST ((Kermany et al., 2018)) and Waterbird ((Sagawa et al., 2019), (Koh et al., 2021)). To create the Cifar10-imb dataset, we followed the settings of (Zhan et al., 2022) and subsampled the training set of Cifar10 with ratios of 1:2:...:10 for classes 0 through 9.

**Baselines:** We compared BRAL-T with three baselines, LossPrediction ((Yoo & Kweon, 2019)), WAAL ((Shui et al., 2020)) and RandomSample. LossPrediction employs an additional module that takes the feature map from the target model as input and predicts the loss for each data point. WAAL adopts min-max loss to better distinguish labeled and unlabeled samples while searching unlabeled batch with higher diversity than labeled samples. According to the experiments in (Zhan et al., 2022), among all the methods, LossPrediction and WAAL achieve best results in 6 benchmarks and competitive results in other 2 benchmarks, therefore we select them as our baselines. For RandomSample, we randomly selected a subset from the unlabeled dataset in each active learning iteration. Besides, for better evaluation, we visualized accuracy-budget curve on Cifar10, Cifar10-imb, Cifar100 and FashionMNIST benchmarks and compared with LossPrediction, WAAL, VAAL ((Sinha et al., 2019)), BADGE ((Ash et al., 2019)), CoreSet ((Zhan et al., 2022)), Cluster-Margin ((Citovsky

Table 1: Experiment results of image classification task on 8 benchmarks.

| Methods | FashionMNIST | | EMNIST | | CIFAR10 | | CIFAR100 | |
|---|---|---|---|---|---|---|---|---|
| | AUBC | F-acc | AUBC | F-acc | AUBC | F-acc | AUBC | F-acc |
| **LossPrediction** | 0.859 | 0.888 | 0.762 | 0.793 | 0.837 | 0.911 | 0.481 | 0.655 |
| **WAAL** | 0.861 | 0.891 | 0.808 | 0.831 | 0.842 | 0.883 | 0.460 | 0.594 |
| **RandomSample** | 0.844 | 0.874 | 0.804 | 0.828 | 0.832 | 0.902 | 0.517 | 0.650 |
| **BRAL-T** | **0.863** | **0.894** | **0.813** | **0.833** | **0.847** | **0.916** | **0.525** | **0.662** |

| Benchmarks | Cifar10-imb | | BreakHis | | Pneum.MNIST | | Waterbird | |
|---|---|---|---|---|---|---|---|---|
| | AUBC | F-acc | AUBC | F-acc | AUBC | F-acc | AUBC | F-acc |
| **LossPrediction** | 0.748 | 0.848 | 0.834 | 0.844 | 0.732 | 0.870 | 0.588 | 0.586 |
| **WAAL** | 0.752 | 0.799 | 0.836 | 0.855 | 0.640 | 0.870 | 0.525 | 0.506 |
| **RandomSample** | 0.710 | 0.810 | 0.834 | 0.832 | 0.706 | 0.652 | 0.586 | 0.502 |
| **BRAL-T** | **0.762** | **0.851** | **0.849** | **0.868** | **0.738** | **0.883** | **0.606** | **0.618** |

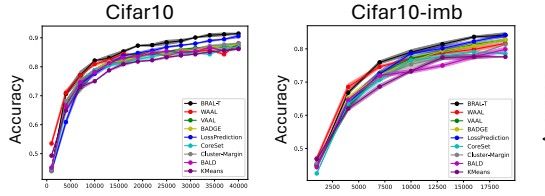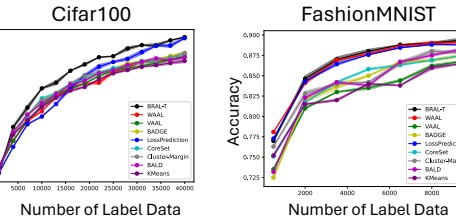

Figure 4: Visualization of experiment results on Cifar10, Cifar10-imb, Cifar100 and FashionMNIST.

et al., 2021)), BALD ((Gal et al., 2017)) and KMeans ((Ash et al., 2019)). To ensure a fair comparison, we used ResNet18 ((He et al., 2016)) as the target model. For more experimental details and hyperparameter settings, please refer to Appendix B.

**Evaluation Metrics:** For all benchmarks, we report evaluation results using two metrics: *area under the budget curve* (AUBC) ((Zhan et al., 2021a), (Zhan et al., 2021b)) and *final accuracy* (F-acc). AUBC refers to the area under the accuracy-budget curve. Methods with a higher AUBC score achieve better overall performance across different sizes of the training set. F-acc refers to the final accuracy achieved after the budget $Q$ is exhausted. The experiments for BRAL-T and the baselines were repeated for 3 trials under different random seeds, and the average of the evaluation results are reported.

**Experiment Results:** The experimental results on 8 benchmarks are presented in Table 1. BRAL-T significantly outperforms RandomSample under all benchmarks. Compared with WAAL and LossPrediction, BRAL-T achieves better AUBC as well as F-acc on all benchmarks. In Figure 4, we visualize the accuracy-budget curves of BRAL-T and baselines on 4 benchmarks. BRAL-T consistently achieves higher accuracy throughout the entire active learning process for Cifar100 and FashionMNIST. In early active learning iterations of Cifar10 and Cifar10-imb, BRAL-T has a bit worse accuracy compared with WAAL, the reasons of which could be attributed to WAAL's emphasis on diversity. However, without adequate consideration for uncertainty cause performance diminishing of WAAL when the size of labeled dataset increases. As comparison, LossPrediction focuses solely on uncertain data with high predicted loss, neglecting the diversity of the selected subset, which results in bad performance in early stage.

## 5.2 ACTIVE LEARNING ON MORE LONG-TAIL DATASETS

**Dataset.** Besides the aforementioned benchmarks, in this section, we focus on long-tail datasets, including CIFAR10-LT and CIFAR100-LT. Both datasets are subsampled from CIFAR datasets and the number of samples within each classes decreases exponentially with factor within 10 and 100. Specifically, we consider 10, 20 and 50 in our experiments. The test images of CIFAR10-LT and CIFAR100-LT are the same as those in CIFAR10 and CIFAR100 datasets respectively. Both the two benchmarks are open-source and can be accessed through huggingface *tomas-gajarsky/cifar10-*

Table 2: Experiment results on Cifar10-LT and Cifar100-LT datasets.

| Methods | Cifar10-LT-r10 | | Cifar10-LT-r20 | | Cifar10-LT-r50 | |
|---|---|---|---|---|---|---|
| | AUBC | F-acc | AUBC | F-acc | AUBC | F-acc |
| SIMILAR | 0.472 | 0.665 | 0.402 | 0.577 | 0.318 | 0.449 |
| LossPrediction | 0.478 | 0.679 | 0.413 | 0.585 | 0.322 | 0.494 |
| WAAL | 0.510 | 0.623 | 0.435 | 0.589 | **0.340** | **0.497** |
| RandomSample | 0.476 | 0.686 | 0.397 | 0.587 | 0.303 | 0.484 |
| BRAL-T | **0.512** | **0.686** | **0.440** | **0.614** | 0.322 | 0.490 |

| Methods | Cifar100-LT-r10 | | Cifar100-LT-r20 | | Cifar100-LT-r50 | |
|---|---|---|---|---|---|---|
| | AUBC | F-acc | AUBC | F-acc | AUBC | F-acc |
| SIMILAR | 0.330 | 0.496 | 0.270 | 0.455 | 0.213 | 0.385 |
| LossPrediction | 0.319 | 0.495 | 0.272 | 0.452 | 0.211 | 0.384 |
| WAAL | 0.297 | 0.454 | 0.249 | 0.397 | 0.198 | 0.340 |
| RandomSample | 0.321 | 0.501 | 0.265 | 0.444 | 0.208 | 0.369 |
| BRAL-T | **0.332** | **0.512** | **0.280** | **0.457** | **0.234** | **0.396** |

Table 3: Experiment results of Active Finetuning task.

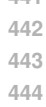
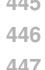

| Methods | Cifar10-imb | | TinyImageNet | |
|---|---|---|---|---|
| | 2% | 3% | 2% | 3% |
| RandomSample | 0.841 | 0.856 | 0.213 | 0.348 |
| ActiveFT | 0.838 | 0.851 | 0.289 | 0.359 |
| BRAL-T | **0.852** | **0.865** | **0.300** | **0.392** |

Table 4: Ablation study result on Cifar10 and Cifar100.

| Baseline | Cifar10 | | Cifar100 | |
|---|---|---|---|---|
| | AUBC | F-acc | AUBC | F-acc |
| PseudoScore | 0.842 | 0.908 | 0.486 | 0.661 |
| BRAL-DiffSet | 0.843 | 0.909 | 0.521 | 0.652 |
| BRAL-T w/o CL | 0.845 | 0.906 | 0.522 | 0.662 |
| RandomSample | 0.832 | 0.902 | 0.517 | 0.650 |
| BRAL-T | **0.847** | **0.916** | **0.525** | **0.662** |

*lt* and *tomas-gajarsky/cifar10-lt*. For all datasets we set the size of initial labeled dataset as 2,000 and the maximum budget to be 20,000. For each active learning iteration, we select 500 samples for oracle to annotate. The training epochs of target model for CIFAR10-LT is set to be 50 and for CIFAR100-LT is 60.

**Experiment Results.** Besides LossPrediction, WAAL and RandomSample, we also compare BRAL-T with SIMILAR ((Kothawade et al., 2021)) which is designed for imbalanced dataset that leverages pseudo label to compute gradient and similar matrix for submodular function. As shown in Table 2, except for CIFAR10-LT-r50 benchmark, BRAL-T achieves the best AUBC as well as F-acc results. As pseudo label is not reliable especially when target model might be overconfident in long-tail dataset, BRAL-T selects informative data with ground-truth label to construct TrustSet which is more reliable to reflect whether target model has sufficiently learnt from related samples. As a result, BRAL-T always performs better than SIMILAR. Moreover, compared with LossPrediction and WAAL, we encourage TrustSet to be balanced which releases the category bias problem in long-tail distribution and contributes to the success of BRAL-T.

## 5.3 ACTIVE LEARNING FOR FINETUNING RESULTS

**Experiment Setting:** (Xie et al., 2023) define the active fine-tuning task as selecting a data subset to fine-tune a pretrained model. For instance, selecting a subset from the Cifar10 dataset to train a classifier pretrained on ImageNet-1k ((Russakovsky et al., 2015)). We adhere to the settings of (Xie et al., 2023) and use Deit-Small ((Touvron et al., 2021)), pretrained with the DINO ((Caron et al., 2021)) framework on ImageNet-1k, as the target model. We chose two datasets for fine-tuning: Cifar10-imb and TinyImageNet ((Le & Yang, 2015)), resizing all images to $224 \times 224$. For more implementation details, we utilize ActiveFT ((Xie et al., 2023)) to select an 1% subset as the initial labeled dataset and select an additional 1% of data for each active learning iteration. The pretrained model is fine-tuned using the SGD optimizer for 1000 epochs with a batch size of 512. Cosine learning rate decay is applied during the fine-tuning phase of each active learning iteration.

**Experiment Results:** Experiments for all methods were repeated across 3 trials, and the average results are reported in Table 3. BRAL-T significantly outperforms the other baselines. ActiveFT aims to select a data subset with a distribution similar to that of the unlabeled data pool. However, if the unlabeled data pool suffers from problems like long-tail distribution or contains a large number of noisy data samples, the selected subset will likely encounter the same issues. For instance, Cifar10-imb is a class-imbalanced dataset, and TinyImageNet includes images that are difficult to classify. Especially for smaller data subsets, only a few data samples from minority categories are included. In contrast, TrustSets are defined to be class-balanced, and curriculum learning encourages the selection of easy or less-noisy data samples in TrustSets. Being trained with the distribution of TrustSets, the policy with the reward function $R_\phi$ learns to select data samples that provide greater benefits for model training.

## 5.4 Ablation Study

**Experiment Setting:** To further evaluate BRAL-T, we conducted ablation studies to demonstrate the benefits of the proposed modules as follows:

- PseudoScore: Instead of training an RL policy, we assign pseudo-labels to the unlabeled data pool based on the category with the highest logit score. We then directly apply curriculum learning and calculate EL2N score based on pseudo-labels. Data subset with top EL2N score will be selected during active learning.

- BRAL-DiffSet: To show the effectiveness of TrustSet, we sort labeled dataset based on the EL2N score and cluster data points into $|L|/|S|$ groups based on the sorted order. Then we select the second-best data group instead of the best one. Other modules remain the same as in BRAL-T.

- BRAL-T w/o CL: For comparison, we remove curriculum learning and directly use the cross-entropy loss function to calculate the EL2N score rather than Super Loss.

**Experiment Results:** For a fair comparison, we use ResNet18 as the target model for all baselines and maintain the same hyperparameters. Table 4 displays the AUBC and F-acc results for the Cifar10 and Cifar100 datasets. BRAL-T surpasses PseudoScore in both AUBC and F-acc. This is because pseudo-labels are often inaccurate, especially in the early stages of active learning where the training set lacks sufficient data to train a high-performance classifier, which can lead to overconfident target model trained on class-imbalanced data. In contrast, selecting the TrustSet based on the labeled dataset is more reliable. Compared to BRAL-DiffSet, BRAL-T also achieves better AUBC and F-acc scores. Since selecting a data group with a lower EL2N score hinders performance, we can empirically prove the correlation between EL2N score and model accuracy. Moreover, the results demonstrate that the RL policy successfully learns to select potentially important data. Curriculum learning also plays a crucial role in the success of BRAL-T, as its removal leads to worse AUBC and F-acc in BRAL-T w/o CL. As mentioned in Section 4.1, curriculum learning aids in selecting easy examples and enhances the performance of the target model in the initial stages.

## 6 Conclusion

In summary, our Reinforcement Learning-based Active Learning framework, BRAL-T, marks a departure from conventional active learning techniques. It leverages TrustSet to more accurately evaluate feature distributions from labeled datasets and employs an RL policy to learn from the selected TrustSet. Unlike methods focusing only on uncertain data, we utilize Super Loss to prioritize easy data samples in early active learning stages. Our reward definition, based on feature distances rather than target model evaluations, simplifies RL training and lowers its complexity. We demonstrate TrustSet extraction using the GradNd score, showing a strong correlation with model accuracy. BRAL-T, benchmarked against LossPrediction, WAAL, and RandomSample across eight image classification tasks, shows superior AUBC and F-acc performance in all cases. Accuracy-budget curves against 8 baselines show benefits of performance of BRAL-T. Moreover, in CIFAR-LT benchmarks, BRAL-T achieves superior preformance compared against baselines, showing its ability to handle long-tail distribution dataset. Additionally, its application in active fine-tuning tasks reveals that BRAL-T surpasses current state-of-the-art results.

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

# A   METHOD DETAILS

BRAL-T comprise two iterative processes: active learning process and reinforcement learning process. Algorithm 1 shows the pseudocode of overall framework. We randomly sampled initial labeled dataset and initialize parameters of target model and reward network in lines 3-5. During the $i$th active learning process (lines 7-9), we trained target model $M_{\theta_i}$ with $i$th labeled dataset $L_i$ from scratch and extract TrustSet $T_i$ from $L_i$, details of which is depicted in Section 4.1. During the reinforcement learning (RL) process (lines 11-17), we followed DQN ((Mnih et al., 2013)) and initialized replay buffer $\mathbb{B}$ to be empty. For each RL iteration, we sample labeled set and unlabeled set from $L_i$ and store state set $\{L'^c_{min}, U'_c, U'^a_c, T_c\}$ into replay buffer (detailed in algorithm 2). To train reward function $R_{\phi_i}$, a data batch is sampled from $\mathbb{B}$ and parameters of $R_{\phi_i}$ is updated based on Eq 10 (detailed in algorithm 3). After the two processes, we sampled a new dataset for oracle to annotate and updated $L_i$ and $U_i$.

---

**Algorithm 1** BRAL-T

---

1: **Input:** Dataset $\mathbb{D}$
2: **Output:** Target Model $M_\theta$
3: Random sample $L_0$ from $\mathbb{D}$ and annotated by oracle;
4: Set $U_0 := \mathbb{D} \setminus L_0$;
5: Initialize $M_{\theta_0}, R_{\phi_0}$;
6: **for** $i = 0$ to $N-1$ **do**
7:     // Active Learning Process
8:     Train $M_{\theta_i}$ with $L_i$ from scratch;
9:     Extract TrustSet $T_i := f(L_i, M_{\theta_i}(L_i))$;
10:
11:     // Reinforcement Learning Process
12:     Initialize Replay Buffer $\mathbb{B}$;
13:     **for** $j = 0$ to $K$ **do**
14:         Sample $L'$ and $U'$ from $L_i$;
15:         Extract set $\{L'^c_{min}, U'_c, U'^a_c, T_c\} = \mathbb{E}(L', U', T_i)$ and store into $\mathbb{B}$; (Algorithm 2)
16:         Sample data from $\mathbb{B}$ and train $R_{\phi_i}$ as Eq 10. (Algorithm 3)
17:     **end for**
18:
19:     // Sample New DataSet
20:     Sample $S_i := \pi(R_{\phi_i}, L_i, U_i)$;
21:     Update $L_{i+1} := L_i \cup S_i$ and $U_{i+1} := U_i \setminus S_i$;
22: **end for**

---

In algorithm 2, we show the pseudocode of data extraction for RL (line 15 of algorithm 1). As illustrated in Section 4.2, we clustered labeled set into $\{L_m\}_{m=1}^M$ and unlabeled set into $\{U_c\}_{c=1}^C$ to formulate state space of RL. For each unlabeled subset, we further cluster $U_c$ into $\{U_c^a\}_{a=1}^{A_c}$ to formulate action space of RL and extract Trustset $T_c$ for each $U_c$. All pairs of $\{L^c_{min}, U_c, U_c^a, T_c\}$ are stored and return as extraction results.

---

**Algorithm 2** Data Extraction For Reinforcement Learning

---

1: **Input:** LabeledSet $L$, UnlabeledSet $U$, TrustSet $T$
2: **Output:** Data list $Out$
3: Initialize output list $Out := []$;
4: Cluster $L$ into $\{L_m\}_{m=1}^M$;
5: Cluster $U$ into $\{U_c\}_{c=1}^C$
6: **for** each $U_c$ **do**
7:     Extract $T_c := T_i \cap U_c$
8:     Cluster $U_c$ into $\{U_c^a\}_{a=1}^{A_c}$;
9:     Calculate $L^c_{min} := \arg\min_m d(L_m, U_c)$;
10:     Store each $\{L^c_{min}, U_c, U_c^a, T_c\}$ into $Out$;
11: **end for**
12: Return $Out$;

---

In algorithm 3, we show the pseudocode of reinforcement learning to train data selection policy (line 16 of algorithm 1). For each gradient step, we sample state and action data from replay buffer $\mathbb{B}$ and extract vector input $S$ and $A$ (line 4-6). Then based on Eq. 8, we calculate the reward for each (state, action) pair as negative distance between data subset and TrustSet (line 7). And based on Eq. 9, we predict reward with current reward function $R_\phi$ (line 8). Finally, we calculate mean square error (MSE) loss between predicted reward $r$ and ground truth reward $R$ and update reward function with gradient descent (line 9).

---

**Algorithm 3** Training of Reinforcement Learning

---

1: **Input:** Replay Buffer $\mathbb{B}$, Reward Function $R_\phi$.
2: **Output:** Update Reward Function $R_\phi$
3: **for** Each Gradient Step **do**
4:     Sample data batch from $\mathbb{B}$ as $\{L_{min}^c, U_c, U_c^a, T_c\}^B$.
5:     Extract state vector input as: $S = [E[L_{min}^c], Var[L_{min}^c], E[U_c], Var[U_c]]$.
6:     Extract action input as: $A = [E[U_c^a], Var[U_c^a]]$.
7:     Calculate reward for each state action pair as Eq. 8: $R = -d(A, T_c)$.
8:     Predicate reward with $R_\phi$ as Eq. 9: $r = R_\phi(S, A)$.
9:     Calculate Loss $L = \text{MSE}(R, r)$ and update $R_\phi$ with gradient descent.
10: **end for**
11: Return $R_\phi$;

---

## B    EXPERIMENT DETAILS

In this section, we introduce more experiment details of Section 5, including architecture of target model we used for image classification and hyperparameter settings of experiments.

| Benchmarks | $|L_0|$ | $|U_0|$ | $Q$ | $b$ | $\#e$ | $C$ |
|---|---|---|---|---|---|---|
| FashionMNIST | 500 | 59,500 | 10,000 | 250 | 40 | 10 |
| EMNIST | 1,000 | 696,932 | 50,000 | 500 | 40 | 62 |
| CIFAR10 | 1,000 | 49,000 | 40,000 | 500 | 50 | 10 |
| CIFAR100 | 1,000 | 49,000 | 40,000 | 500 | 60 | 100 |
| CIFAR10-imb | 1,000 | 27,239 | 20,000 | 500 | 50 | 10 |
| CIFAR10-LT | 2,000 | - | 20,000 | 500 | 50 | 10 |
| CIFAR100-LT | 2,000 | - | 20,000 | 500 | 60 | 100 |
| BreakHis | 100 | 5,436 | 5,000 | 100 | 30 | 2 |
| PneumoniaMNIST | 100 | 5,132 | 5,000 | 100 | 30 | 2 |
| Waterbird | 100 | 4,695 | 4,000 | 100 | 30 | 2 |

Table 5: Setting of benchmarks. Where $|L_0|$ refers to size of initial labeled set, $|U_0|$ refers to size of initial unlabeled data pool, $Q$ refers to budget, $b$ refers to batch size for target model training, $\#e$ refers to number of epoch for target model training and $C$ refers to number of clusters from unlabeled data pool. For all the benchmarks, the number of clusters $M$ from labeled dataset is set to be the same as $C$ and number of candidate action for $U_c$ is set to be 5.

**DataSets.** We evaluated BRAL-T on the image classification task across 5 benchmarks, including Cifar10, Cifar100 ((Krizhevsky et al., 2009)), Cifar10-imb, EMNIST ((Cohen et al., 2017)), and FashionMNIST ((Xiao et al., 2017)). To create the Cifar10-imb dataset, we followed the settings of (Zhan et al., 2022) and subsampled the training set with ratios of 1:2:...:10 for classes 0 through 9. We also evaluated our framework on medical imaging analysis tasks across 2 benchmarks, including Breast cancer Histopathological Image Classification (BreakHis) ((Spanhol et al., 2015)) and Chest X-Ray Pneumonia classification (Pneumonia-MNIST) ((Kermany et al., 2018)). Additionally, we assessed our framework on an object recognition dataset with correlated backgrounds (Waterbird) ((Sagawa et al., 2019), (Koh et al., 2021)), which contains waterbird and landbird classes manually mixed with water and land backgrounds. To further evaluate BRAL-T on long-tail datasets, we also consider CIFAR10-LT and CIFAR100-LT where the number of samples within each classes decreases exponentially with factor to be 10, 20 or 50.

The detail setting for each benchmark are shown in Table 5, including initial data size of labeled dataset $|L_0|$ and unlabeled dataset $|U_0|$, final budget $Q$ of labeled dataset, batch size $b$ for data subset selection in each active learning iteration, training epoch $\#e$ for target model training, and category number $C$ for dataset.

**Model Details.** Following the setting of (Zhan et al., 2022), we use Resnet18 ((He et al., 2016)) as the target model for image classification tasks. For Cifar10, Cifar10-imb, Cifar100 and PneumoniaMNIST, we replaced the kernal size of first convolutional layer to be $3 \times 3$ and stride to be 1 in order to handle image with smaller size. For grayscale images such as FashionMNIST and EMNIST datasets, we add an additional convolutional layer before the first layer of Resnet with $1 \times 1$ kernal to increase the channel number of images to be 3. Furthermore, we trained the target models of all baselines for the same number of epochs, as shown in Table 5. For LossPrediction, the target model is trained with both classification loss and loss prediction loss for the first 20 epochs. After 20 epochs, only the gradient from the classification loss is back-propagated through the target model.

**Model and Hyperparameters Setting:** We constructed the reward function $R_\phi$ using a fully connected network comprising 2 hidden layers, each with 512 units, and use the ReLU activation function. SGD was employed as the optimizer for $R_\phi$, with the learning rate set at 0.01. For hyperparameters of curriculum learning, we follow the setting of SuperLoss ((Castells et al., 2020)) and set $\tau = \log |K|$ where $|K|$ is the category number. Additionally, we set the value of $\lambda$ to be 0.25 for EMNIST, CIFAR100 and TinyImageNet datasets and 1.0 for the others. During active learning, We train target model with SGD optimizer for PneumoniaMNIST and Waterbird benchmarks and Adam optimizer for other datasets.

After each active learning iteration, we sampled 30 pairs of $L'$ and $U'$ from the existing labeled set $L$ to train the policy, setting the batch size to 100 pairs of state, action, and reward. Following each sampling, we trained $R_\phi$ for 20 iterations, resulting in a total of 600 iterations for the entire RL training process. As shown in Table 5, the number of clusters $C$ for unlabeled set and $M$ for labeled set are set to be the same as category number for related benchmark. And the number of candidate action $A_c$ for each unlabeled cluster $U_c$ is set to be 5 during the experiment.

# C  MORE EXPERIMENT RESULTS

In this section, we introduce more experiments and results. First of all, we show the confidence interval results for Table 1 over 8 benchmarks in C.1. Then we evaluate BRAL-T by calculating penalty matrix in C.2. Moreover, to show the efficiency of BRAL-T, we compare time overhead between BRAL-T and baselines in C.3. Finally, in C.4, we show more ablation studies of BRAL-T.

## C.1  CONFIDENCE INTERVALS OF RESULTS IN IMAGE CLASSIFICATION TASKS.

Besides representing average value of AUBC and F-acc of BRAL-T and baselines on image classification benchmarks, Table 6 shows the confidence interval of experiment results. In general, BRAL-T results are stable and robust over different experiment trials.

| Methods | FashionMNIST | | EMNIST | | CIFAR10 | | CIFAR100 | |
|---|---|---|---|---|---|---|---|---|
| | AUBC | F-acc | AUBC | F-acc | AUBC | F-acc | AUBC | F-acc |
| LossPrediction | ±0.002 | ±0.038 | ± 0.016 | ± 0.022 | ±0.006 | ± 0.012 | ±0.019 | ±0.012 |
| WAAL | ±0.002 | ±0.015 | ± 0.012 | ±0.015 | ± 0.006 | ±0.009 | ±0.006 | ±0.011 |
| RandomSample | ±0.001 | ±0.009 | ± 0.004 | ±0.007 | ±0.003 | ± 0.011 | ±0.003 | ±0.008 |
| BRAL-T | ±0.001 | ±0.008 | ±0.005 | ±0.014 | ±0.003 | ±0.006 | ±0.004 | ±0.009 |

| Benchmarks | Cifar10-imb | | BreakHis | | Pneum.MNIST | | Waterbird | |
|---|---|---|---|---|---|---|---|---|
| | AUBC | F-acc | AUBC | F-acc | AUBC | F-acc | AUBC | F-acc |
| LossPrediction | ±0.011 | ±0.017 | ±0.026 | ±0.037 | ±0.023 | ±0.038 | ±0.014 | ±0.097 |
| WAAL | ±0.008 | ±0.013 | ±0.016 | ±0.042 | ±0.018 | ±0.021 | ±0.011 | ±0.078 |
| RandomSample | ± 0.013 | ± 0.019 | ±0.015 | ±0.050 | ±0.001 | ±0.009 | ±0.005 | ±0.059 |
| BRAL-T | ± 0.012 | ± 0.008 | ±0.017 | ±0.037 | ±0.012 | ±0.013 | ±0.007 | ±0.024 |

Table 6: Confidence Interval of Experiment results of image classification task.

## C.2 PAIRWISE COMPARISON

We further compare BRAL-T with VAAL ((Sinha et al., 2019)), SAAL ((Kim et al., 2023)) and BAIT ((Ash et al., 2021)) on Cifar10, Cifar10-imb and FashionMNIST datasets by pairwise penalty matrix following (Ash et al., 2021). For each benchmark, we collect accuracy results achieved by all baselines. For pairwise comparison between the method for $i$th row ($r_i$) and the method in $j$th column ($c_j$), we add a score to element $e_{ij}$ whenever $r_i$ achieves better accuracy result in one budget of data subset for a benchmark, which means the better $r_i$ performs compared with $c_j$, the higher score $e_{ij}$ will be.

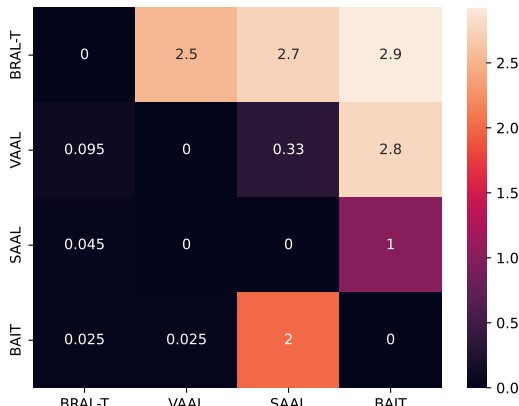

Figure 5: Pairwise Comparison of BRAL-T, VAAL, SAAL and BAIT.

Figure 5 represents the pairwise comparison results. Compared with all baselines, BRAL-T achieves highest value in $e_{\text{BRAL-T},\cdot}$ and lowest value in $e_{\cdot,\text{BRAL-T}}$.

## C.3 TIME OVERHEAD COMPARISON

To evaluate the efficiency of BRAL-T, we compare the time overhead with LossPrediction, WAAL, VAAL and SIMILAR on Cifar10 and Cifar100 datasets. All experiments were conducted using a single Quadro RTS 6000 GPU core with CUDA Version 11.4, and the hardware setup included a 64-core Intel Xeon Gold 5218 CPU. Figure 6 shows the time cost results along with active learning iteration.

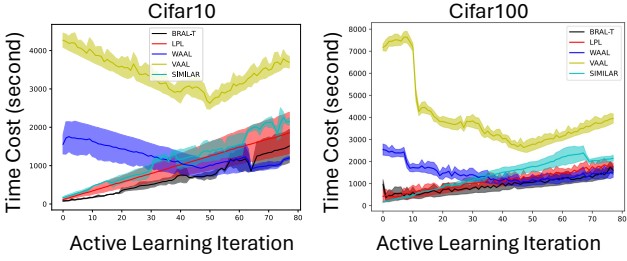

Figure 6: Time Cost.

The time cost associated with BRAL-T increases with each active learning iteration as the labeled set expands and more data samples are clustered during the reinforcement learning process. However, compared to other baselines, BRAL-T consistently demonstrates efficiency, maintaining a competitive edge in terms of computational resource utilization.

**Agent Reuse.** A potential way to further improve the efficiency of BRAL-T is reusing RL agent for all active learning iterations. However, considering the distribution shift of labeled dataset, distribution of TrustSet will also shift during active learning. For this reason, we apply two RL agents during active learning, one of which is trained in the first active learning step and remains

unchanged for early active learning iterations; the other one of which is maintained for the rest iterations. Specifically for CIFAR10-imb dataset, we use the first agent for the first 20 iterations and the second agent for the rest 20 iterations. The result is shown in Table 7 below: where BRAL-T

| Method | AUBC | F-Acc |
|---|---|---|
| LossPrediction | 0.748 | 0.848 |
| WAAL | 0.752 | 0.799 |
| RandomSample | 0.710 | 0.810 |
| BRAL-T | **0.762** | **0.851** |
| BRAL-T (two agents) | 0.755 | 0.837 |

Table 7: BRAL-T reusing two RL agents.

with agent reusing surprisingly achieves better AUBC results compares with other baselines. With a more careful separation of active learning stages and RL agents, we believe the performance could be further improved.

## C.4 MORE ABLATION STUDY

To evaluate the robustness of BRAL-T, we run BRAL-T on Cifar10-imb dataset under different qualifies of initial labeled dataset. Moreover, we show the performance of BRAL-T with different candidate action numbers. To evaluate the quality of RL approximation, we apply ground truth labels for TrustSet selection and compare the accuracy results with BRAL-T.

**Quality Effect of Initial Labeled Set.** We explored the impact of the initial labeled set's quality by applying three different sampling methods to construct the initial labeled set from the Cifar10-imb dataset:

- **Random Sample**: We randomly sample data from the unlabeled pool to form the initial labeled set which maintains a similar category distribution with unlabeled pool.
- **Twisted Main**: We sort the 10 categories by the number of data samples first and then select 50 samples from 5 rare classes and 950 samples randomly from the other 5 main classes.
- **Twisted Rare**: Similar to Twisted Main, we randomly select 50 samples from 5 main classes and 950 samples from the other 5 rare classes.

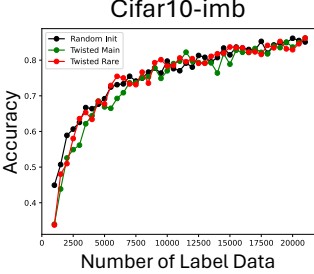

Figure 7: Accuracy-budget curve of Different Initial Labeled Set Quality.

| Initial Method | AUBC | F-Acc |
|---|---|---|
| **Random** | 0.762 | 0.851 |
| **Twisted Main** | 0.750 | 0.855 |
| **Twisted Rare** | 0.756 | 0.855 |

Table 8: Experiment Result of Different Initial Labeled Set Quality.

The results, depicted in the Figure 7, indicate that BRAL-T's performance varies with the quality of the initial labeled set, particularly when labeled data is scarce. However, as the size of the labeled dataset increases, the accuracy differences become negligible, demonstrating BRAL-T's robustness to the initial set's composition. Despite the initial set's quality impacting BRAL-T's performance, in Table 8, the AUBC results in the twisted cases are competitive with the results of the WAAL baseline in Table-2, and all achieve better F-Acc compared with other baselines.

**Ablation Study on Different Action Numbers.** In the reinforcement learning process, we set number of candidate action to be 5 in Section 5. To evaluate the impact of varying action space sizes,

we conducted an ablation study on the Cifar10-imb dataset, comparing BRAL-T's performance across different numbers of actions: 5, 10, 50, and 100. The results are shown in Table 9

| # Actions | AUBC | F-Acc |
|---|---|---|
| 5 | 0.762 | 0.851 |
| 10 | 0.763 | 0.853 |
| 50 | 0.755 | 0.851 |
| 100 | 0.758 | 0.854 |

Table 9: Ablation Study on Different Candidate Action Number.

Under all different setting of action numbers, BRAL-T achieves best AUBC and F-Acc results compared with baselines in Table 1. Setting a large number of actions will increase the complexity of policy training. As we keep the policy architecture to be the same and simple for time efficiency, in some active learning iteration policy might not be trained well with large action number which lead to a small drop of AUBC score. But in general, our method is robust to action number. The reason we choose 5 in the experiment is mainly for the consideration of time efficiency.

**Ablation Study on Different Setting of $\lambda$.** During the TrustSet extraction, we introduce curriculum learning where $\lambda$ is introduced to control the effect of SuperLoss. We study the impact of $\lambda$ on the CIFAR10-imb dataset for further sensitivity analysis, the result is shown in table 10 below:

| $\lambda$ | AUBC | F-Acc |
|---|---|---|
| 0.25 | **0.762** | **0.851** |
| 1.00 | 0.752 | 0.842 |
| 2.00 | 0.749 | 0.830 |

Table 10: Impact of $\lambda$ value on CIFAR10-imb dataset.

Increasing the value of $\lambda$ reduces the influence of SuperLoss on the task loss. In an imbalanced dataset, data samples are limited, especially in rare classes. Focusing on difficult data during the early stages of active learning can significantly increase the difficulty of model training. As a result, increasing $\lambda$ leads to a reduction in AUBC and F-Acc for BRAL-T, highlighting the importance of incorporating curriculum learning into the active learning process. However, overall, the AUBC and F-Acc values remain competitive with the baselines presented in Table 1 of the paper.

**Compare between RL and Ground Truth Labels.** Although label information of unlabeled data pool is not available during active learning, in order to evaluate the approximation performance of RL policy, for baseline GradNd we assume ground truth label of unlabeled data pool is available when calculating the GradNd score of data samples and we pick class-balanced data with top GradNd score for each active learning iteration. We compare BRAL-T with GradNd on Cifar10 and Cifar10-imb datasets and shows the accuracy-budget results in Figure 8.

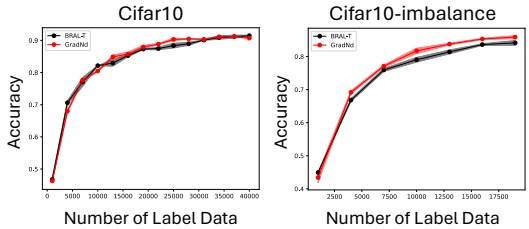

Figure 8: Comparison between BRAL-T with RL policy and Ground Truth Labels.

In Cifar10 dataset, BRAL-T achieves good performance to approximate TrustSet, where only small gap exists when labeled dataset becomes larger. In Cifar10-imb dataset, similarly, when labeled dataset is limited, BRAL-T achieves similar accuracy compared with GradNd. When the size of labeled dataset becomes larger, the accuracy difference performs to be acceptable larger. As a conclusion, the RL policy in BRAL-T achieves good performance to approximate ground truth TrustSet selection.

