# OpenReview forum: "Labeled TrustSet Guided: Combining Batch Active Learning with Reinforcement Learning"
_ICLR.cc/2025/Conference — ICLR 2025 Conference Withdrawn Submission_

### Official Review · Reviewer_g1aP · 2024-10-27

**Soundness:** 2
**Presentation:** 2
**Contribution:** 1
**Rating:** 3
**Confidence:** 5

**Summary:**

This paper proposed an RL-based method to address a classical setting, i.e., active learning. Specifically, this paper designed the reward function, state space, and action space to help the sample query stage. The experimental results were conducted among 5 benchmarks, validating its effectiveness.

**Strengths:**

1. It is a good trial to adopt the concept of reinforcement learning to help address batch active learning.
2. This paper is structured, and well-written.

**Weaknesses:**

1. Insufficient Experimental Support on the Design of Curriculum Learning Mechanism When learning the TrustSet, this paper adopts some concepts from curriculum learning, i.e., super loss, to help address the uncertainty of the selected samples, which stands strongly on an argument "...Data with high GradNd scores tend to be difficult and uncertain samples...", and this paper uses the results in Figure 3 to claim the rightness of this conjecture. However, compare to the version in CVPR’24, the authors delete the pilot experimental parts, (Figure 3) in original version, which further weakens the claim since no pilot experiment is built. Most importantly, we are not clear whether this issue only occurred for the GradNd-based methods since only this one is shown (What if other methods, like uncertainty score, BatchBald, GrandMatching, Submodular,..., have no such issue?), which I believe needs more comparing methods to help support such motivation to use the SuperLoss.

2. Unclear performance bound about the TrustSet. The aim of this paper is to construct and optimize the selected samples, treating them as the best suited during each query. This also suggests that using all samples could be the best. Therefore, there should be a theoretical performance bound about the proposed RL method, which aims to approximate the optimal "TrustSet". Besides, does TrustSet have to be built by GraNd? Why not other methods? This paper needs to make a clear illustration for that.
The whole design of RL framework is built for the cluster-wise groups instead of each sample, which is obviously different from those traditional methods focusing on the sample score. It also means that the proposed RL method is only applicable to the group-level selection, which, intuitively, may not be effective when the number of samples in the unlablled pool is small (since each sample counts).

3. I do not see clear novelty and value of such RL-based design. As this paper claims in Related Work about "Active Learning with RL", the deficiency of some methods using accuracy metrics are " ...the relationship between the target model’s predictions and the training set is complex, making it difficult to train the RL policy..." However, the reward function in the proposed framework is not based on an intuitive evaluation criteria but a TrustSet, which is not as measurable as the former. So how such a design could be more "simple" than those criteria-based methods to train the RL policy? Besides, the most important of RL is the design of the reward function, but I cannot see the clear value and novelty of designing such this reward function.

4. Based on the results in Figure 4, this method works not as promising as expected. Besides, where is the results of Tiny-ImageNet with progressive data volume?not 2%-3%

5. The time complexity analysis of this RL-based method lacks comparison to other methods, and the experimental results about that shall be presented (such as training time, FLOPS...)

**Questions:**

No

---

### Official Review · Reviewer_kfiz · 2024-10-27

**Soundness:** 2
**Presentation:** 1
**Contribution:** 1
**Rating:** 3
**Confidence:** 3

**Summary:**

This paper proposes the Batch Reinforcement Active Learning with TrustSet (BRAL-T) ensuring a class-balanced sampling for long-tail problem. By introducing RL in batch active learning scenario, TrustSet selects high-quality samples from the unlabeled data. Extensive experiments demonstrate the effectiveness of the proposed BRAL-T.

**Strengths:**

1. Defining state space with $L$ and $U$ is novel.
2. Extensive experiments on long-tail datasets are interesting.

**Weaknesses:**

1. Understanding and reading research papers is quite challenging. The notation should be well-defined in Section 3.
2. TrustSet naively incorporates the concepts of class-balanced and curriculum learning into the existing methods such as GradNd and Super Loss.
3. There are cases where a subset $S$ is subsampled from $L$. I’m curious about the intuition behind this approach and why the entire set $L$ isn’t used.

**Questions:**

Please see “weaknesses” part.

---

### Official Review · Reviewer_2Juo · 2024-11-03

**Soundness:** 2
**Presentation:** 2
**Contribution:** 3
**Rating:** 5
**Confidence:** 3

**Summary:**

The paper presents Batch Reinforcement Active Learning with TrustSet (BRAL-T) framework, which combines batch active learning (BAL) and reinforcement learning (RL). This method introduces TrustSet, which selects a balanced subset of labeled data that improves model performance, especially for data with long-tail distribution. To adapt TrustSet for unlabeled data, BRAL-T uses a Reinforcement Learning (RL) sampling policy trained on the labeled set to select unlabeled samples approximating the qualities of TrustSet data. The authors validate the performance of BRAL-T across several image classification and fine-tuning benchmarks.

**Strengths:**

* The paper introduces a novel TrustSet approach designed to balance class distribution and mitigate the long-tail problem present in CoreSet. In addition, incorporating reinforcement learning to extend the properties of TrustSet, based on labeled data, to unlabeled data is a promising direction in active learning.

* The authors perform comprehensive experiments across eight active learning benchmarks and various long-tailed active learning / fine-tuning tasks. The authors also perform rigorous baseline comparisons and ablation studies, demonstrating that each component of the framework contributes meaningfully to performance improvements.

* The presentation is overall well-organized, with detailed figures and algorithms that illustrate each component of the proposed methods.

**Weaknesses:**

* To my understanding, TrustSet is a pruning/selecting strategy for **labeled data**, and the benefits highlighted in the abstract "selects the most informative data from the labeled dataset, ensuring a balanced class distribution to mitigate the long-tail problem" and "optimizes the model’s performance by pruning redundant data and using label information to refine the selection process" (L018-L022) both apply to the **labeled data**. However, the authors do not provide theoretical proof or experimental results to substantiate these claims regarding the **labeled data**.
  I acknowledge the authors provide some discussion about BRAL-DiffSet in Section 5.4. However, in my personal opinion, this ablation study validates the effectiveness of EL2N score on active learning, rather than the effectiveness of TrustSet on the selection of labeled data. Given that TrustSet is a key contribution of this work, I believe it is crucial to validate its effectiveness through empirical evidence or theoretical analysis.

* Setting the number of candidate actions $A_c$ to a fixed number might be an issue, particularly for imbalance/long-tail datasets. In particular, if the clustering divided the unlabeled $U$ into $C$ as expected with each cluster primarily containing samples from the same class, the resulting $c$ clusters would be highly imbalanced. In such case, setting $A_c$ to a fixed number will make the sub-clusters, i.e., the candidate actions, $U_c^a$ to be imbalanced as well across $c$ clusters, which seems to contradict the objective of achieving balanced data selection. Can the authors elaborate on why choosing a fixed value for $A_c$?

* It seems that the experiments do not contain any RL-based active learning baselines. Could the authors elaborate on why TAILOR [1] is not included, as it also aims to find class-balanced examples in active learning by incorporating RL?

* Some presentation issues. For example, most citations in Section 2 and Section 5 are in double brackets; The legends in Figure 4 are barely readable; The captions are above tables in the main manuscript but are below tables in the Appendix; the baseline methods WAAL, LossPrediction, and SIMILAR are not discussed in Related Works.

**Questions:**

* I find some of the experimental settings to be confusing. As stated in Appendix B, the active learning experiments involved querying nearly the entire unlabeled datasets—for instance, 5,000 out of 5,436 for BreakHis, 5,000 out of 5,132 for PneumoniaMNIST, and 4,000 out of 4,695 for Waterbird. This large query budget effectively undermines the purpose of active learning. I understand these settings follow [1], but I would appreciate it if the authors could explain their rationale for adopting such an approach.

* In my humble opinion, Figure 1 seems to be a generic framework of active learning with RL, and does not provide much insight for the proposed BRAL-T method. I would suggest the author add more information to Figure 1 or merge it with Figure 2.

* I would kindly suggest the authors double-check their citations. For example, CoreSet is referred to [2] instead of the original paper, and both BADGE and KMeans refer to the same paper [3].


[2] Zhan, Xueying, et al. "A comparative survey of deep active learning." arXiv preprint arXiv:2203.13450 (2022).
[3] Ash, Jordan T., et al. "Deep batch active learning by diverse, uncertain gradient lower bounds." arXiv preprint arXiv:1906.03671 (2019).

---

### Official Review · Reviewer_4X88 · 2024-11-10

**Soundness:** 3
**Presentation:** 3
**Contribution:** 3
**Rating:** 6
**Confidence:** 4

**Summary:**

The paper proposes a data selection method named as TrustSet which takes into account uncertainty, diversity and class distribution. TrustSet in combination with reinforcement learning based sampling policy introduced BRAL-T framework which extends the benefits of TrustSet to select meaningful data.

**Strengths:**

The BRAL-T framework is quite effective considering good performance on some standard datasets. Overall, the paper is easy to follow and well explained.

**Weaknesses:**

It’s not clear why partcularly GradNd score is used for TrustSet extraction in case of large datasets and complex models. GranND scores appear to be highly sensitive towards small changes in model parameters and may also add high computational overhead.

Perhaps large datasets such as ImageNet could be used to test the performance and time complexity of the proposed method in an image classification setup.

It’d be interesting to understand the motivation to select negative Wasserstein distance as reward function since it’s considerably computationally expensive and might turn out to be challenging in high-dimensional spaces. It’d also be useful to compare the proposed framework to a few recent baselines as well.

**Questions:**

Please refer to the Weaknesses section.

---

### Note · Authors · 2024-11-14

I have read and agree with the venue's withdrawal policy on behalf of myself and my co-authors.